# Expression of Angiopoetin-Like Protein-4 and Kidney Injury Molecule-1 as Preliminary Diagnostic Markers for Diabetes-Related Kidney Disease: A Single Center-Based Cross-Sectional Study

**DOI:** 10.3390/jpm13040577

**Published:** 2023-03-24

**Authors:** Gulnaz Bano, Mohammad Tarique Imam, Ram Bajpai, Ghada Alem, Varun Kumar Kashyap, Anwar Habib, Abul Kalam Najmi

**Affiliations:** 1Department of Pharmacology, School of Pharmaceutical Education and Research, Jamia Hamdard, New Delhi 110062, India; 2Department of Clinical Pharmacy, College of Pharmacy, Prince Sattam Bin Abdul Aziz University, Al-Kharj 11942, Saudi Arabia; 3School of Medicine, Keele University, Staffordshire ST5 5BG, UK; 4Department of Community Medicine, Hamdard Institute of Medical Sciences and Research, Jamia Hamdard, New Delhi 10062, India; 5Department of Medicine, Hamdard Institute of Medical Sciences and Research, Jamia Hamdard, New Delhi 10062, India

**Keywords:** biomarkers, end-stage renal disease, macroalbuminuria, microalbuminuria, type 2 diabetes mellitus

## Abstract

The purpose of the study was to examine the urinary levels of kidney injury molecule-1 (KIM-1) and angiopoietin-like protein-4 (ANGPTL-4) in individuals with diabetic kidney disease (DKD) and their association with established DKD diagnostic markers such as albuminuria and estimated glomerular filtration rate (eGFR). Levels of ANGPTL-4 and KIM-1 were estimated in urine samples. A total of 135 participants were recruited into three groups: 45 diabetes type 2 patients in the control group and 90 DKD patients in two disease groups. Concentrations of ANGPTL-4 and KIM-1 were conclusively related to the urinary albumin–creatinine ratio (UACR). Also, the levels of both ANGPTL-4 and KIM-1 were negatively associated with the eGFR. Multivariable Poisson regression analysis showed that urinary ANGPTL-4 (PR: 3.40; 95% CI: 2.32 to 4.98; *p* < 0.001) and KIM-1 (PR: 1.25; 95% CI: 1.14 to 1.38; *p* < 0.001) were prevalent in DKD patients. Receiver operating characteristic (ROC) analysis of urinary ANGPTL-4 and KIM-1 in the combined form resulted in an area under curve (AUC) of 0.967 (95%CI: 0.932–1.000; *p* < 0.0001) in the microalbuminuria group and 1 (95%CI: 1.000–1.000; *p* < 0.0001) in the macroalbuminuria group. The association of urinary levels of ANGPTL-4 and KIM-1 with UACR and eGFR and significant prevalence in the diabetic kidney disease population illustrates the diagnostic potential of these biomarkers.

## 1. Introduction

One of the most significant microvascular consequences of diabetes is diabetic kidney disease (DKD) [1]. About 20 to 40 percent of all people with type 1 or type 2 diabetes go on to develop DKD [2]. It quietly outsets without any significant clinical signs at the primary stage and progresses to end-stage renal disease [3]. Pathologically DKD is characterized by ultra-structural damages in the nephron including thickening of the glomerular and tubular basement membrane, accumulation of extracellular matrix, expansion of the mesangial, hypertrophy of the glomerulus and tubule, rearrangement of the podocyte’s cytoskeleton, and tubular cell damage [4,5]. Persistent proteinuria (micro- or macroalbuminuria) on at least two occasions, spaced three to six months apart, is a clinical hallmark of DKD. It is still the gold standard for determining if a patient has overt nephropathy or diabetic renal failure [6]. However, it has several limitations, and there is promising evidence that diabetic patients may already be suffering from advanced glomerular lesions even in a nonalbuminuria state [7]. Asserting albuminuria as a surrogate marker of kidney disease progression would be erroneous, as it poses uncertainty in disease prediction [8]. An expeditious diagnosis and specific assessment of DKD at an early stage are critical to the management of the disease. This is apparent from the research that some of the markers associated with podocyte dysfunction, such as angiopoietin-like protein-4 (ANGPTL-4) and tubular injury markers like kidney injury molecule-1 (KIM-1), may be successfully utilized to diagnose DKD at the early stage [9,10].

ANGPTL-4, a 50 kDa protein is secreted by a variety of cells and belongs to the family of angiopoietin-like proteins such as adipocytes, liver and endothelial cells, cardiomyocytes, and macrophages [11]. Recently, researchers have discovered the association of podocyte dysfunction with the expression of ANGPTL-4 [12]. It was found to be upregulated in certain podocyte diseases, such as membranous nephropathy and experimental minimal change disease. Further, ANGPTL-4 expressed in podocytes was found to be related to increased albuminuria, suggesting a role of ANGPTL-4 in overt proteinuria in glomerular diseases [13,14].

KIM-1 is a glycoprotein, which is a type 1 cell membrane, manifesting on proximal tubule cells in the apical membrane, and being unstable, its ectodomain splits, sheds into the lumen of the tubule, and is excreted into the urine [10]. The expression of KIM-1 is extremely specific and sensitive to kidney tubular injury; it is undetectable in the normal kidney but significantly upregulated in acute injury or an inflamed state of the kidney [15].

The purpose of the current study was to examine the implication of the combination of two distinct site markers, urinary ANGPTL-4 and urinary KIM-1, in the diagnosis of DKD. The existing evidence on ANGPTL-4 and KIM-1 as biomarkers of diabetes-related kidney damage among Indians is limited and consequently requires further investigation.

## 2. Materials and Methods

### 2.1. Study Design and Participants

At Jamia Hamdard’s Hakeem Abdul Hameed Centenary (HAHC) Hospital in New Delhi, this cross-sectional study was carried out from January 2018 to March 2020. Patients were enrolled in the study who met the inclusion and exclusion criteria and were categorized into three age-matched groups. The control group (normoalbuminuria group, *n* = 45, Appendix A) included type 2 diabetes mellitus (T2DM) patients with no kidney disease and urinary albumin–creatinine ratio (UACR) values within the normal range (<30 mg/g). Microalbuminuria group (*n* = 45, Appendix A) included DKD patients with moderately increased UACR (30–300 mg/g), and the macroalbuminuria group (*n* = 45, Appendix A) included DKD patients with severely increased UACR (>300 mg/g). The inclusion criteria were the following: (a) Indian nationals of either sex with age >30 years, (b) confirmed T2DM, diagnosis based on either a fasting plasma glucose (FPG) level ≥126 mg/dl or a 2 h plasma glucose level ≥200 mg/dl or glycosylated hemoglobin (HbA1c) level of 6.5% or higher as per the American Diabetes Association (ADA) guidelines, and (c) established DKD diagnosed as per KDIGO clinical practice guidelines characterized by a persistent increase in UACR (>30 mg/g) and/or a decline in the estimated glomerular filtration rate (eGFR) below 60 mL/min per 1.73 m^2^ for the micro- and microalbuminuria groups [16,17]. The exclusion criteria were (a) a non-willingness to participate in the study; (b) the presence of other renal or urinary tract illness confirmed by clinical or laboratory evidence, cerebrovascular disease, inflammatory diseases, infectious disease, cancer, tumor, or recent surgery; and (c) pregnancy. The institutional ethics committee of Jamia Hamdard approved the study protocol (approval number: JHIE-2017-11/20; 14/17). Written informed consent was acquired from the participants, and the study was conducted in accordance with the Declaration of Helsinki [18].

### 2.2. Sample Size

The estimation of the sample size was based on the prevalence of diabetic kidney disease [19,20]. It was determined by utilizing the following formula: N = (Z21-α/2 p(1-p))/d2. The calculated sample size was 37 where ‘p’ prevalence or proportion of event was taken as 2.5%, and ‘d’ absolute error was 5% with 95% α. However, 20% was taken as non-response; hence, the estimated sample size was 44 in each group. A total of 182 patients were screened, and 135 patients, 45 in each group, were recruited in the study, as shown in the Figure 1.

### 2.3. Data Collection

After patient enrollment in the study, data were collected from the participants. Demographic details including age, gender, weight, height, marital status, substance use (like smoking), education, employment, duration, and family history of T2DM were recorded in a validated and approved case recording form.

### 2.4. Sample Collection and Biochemical Parameters Assessment

Spot urine and blood samples were collected from the participants. Blood samples were centrifuged at 4500 rpm at 4.0 °C for 10 min, and separated plasma was stored at −80.0 °C until further analyses. Collected urine samples were centrifuged in a refrigerated centrifuge at 3500 rpm and aliquoted at −80.0 °C for further analyses. FPG was quantified with spectrophotometric (Roche Diagnostics, Mannheim, Germany) using the glucose oxidase and peroxidase (GOD-POD, Prague, Czech Republic) method. HbA1c was assessed using fully automated high-performance liquid chromatography (Bio-Rad D10, Hercules, CA, USA), and values are reported in percentages as per the National Glycohemoglobin Standardization Programme (NGSP, Holly Springs, GA, USA) recommendations. Serum albumin was estimated with the bromocresol green method (Beckman Coulter AU 480, Brea, CA, USA), and the albumin–globulin ratio was derived from albumin and globulin values. Protein total was determined with the biuret method, and serum creatinine and urine creatinine were analyzed with the Kinetic Jaffe method using an automatic analyzer (Beckman Coulter AU 480). The chronic renal disease epidemiology collaboration equation was used to determine eGFR (CKD-EPI Creatinine Equation), alkaline phosphatase was estimated with the modified International Federation of Clinical Chemistry (IFCC) method, and total cholesterol was determined by the enzymatic method using an automatic analyzer (Beckman Coulter AU 480) [21]. Uric acid concentrations were evaluated with the uricase peroxidase method, blood urea nitrogen (BUN) with kinetic ultra violet assay, and calcium with Arsenazo III method using an automatic analyzer (Beckman Coulter AU 480). Sodium (Na), potassium (K), chloride (Cl), and phosphorus (P) levels were determined via ion selective electrode (ISE) measurement using a fully automated analyzer (Roche 9180 Electrolyte Analyzer). Urinary albumin creatinine ratio was derived from the values of urine albumin and urine creatinine. Urinary concentrations of ANGPTL-4 and KIM-1 were determined using enzyme-linked immunosorbent assay (ELISA) kits {RayBiotech (Norcross, GA, USA), Genexbio (Delhi, India) respectively}. The detection range of the ANGTPL-4 ELISA kit was 20 pg/mL–20,000 pg/mL with a sensitivity of 20 pg/mL, the intra-assay coefficient of variation (CV) was <10%, and the inter-assay CV was <12%. The KIM-1 ELISA kit had a detection range of 0.05 ng/mL to 10 ng/mL and a sensitivity of 0.01 ng/mL. There were 8% and 10% intra- and interassay CVs, respectively.

### 2.5. Statistical Analysis

The mean and standard deviation (SD) are used in descriptive statistics to represent quantitative parameters, where as absolute numbers and corresponding percentages are used to represent categorical data. The Kolmogorov–Smirnov test was used to determine whether the data were normal. The Bonferroni correction (P ≤ α/n) after the one-way analysis of variance (one-way ANOVA) test in multiple comparisons was used for testing the statistical differences among the quantitative parameters between the groups. We performed the chi-square test to determine the correlation for categorical data. The correlations between urinary concentrations of ANGPTL-4 and KIM-1 with other metabolic parameters were examined using Spearman’s rank-order correlation.

The association of effect of urinary levels of ANGPTL-4 and KIM-1 and other potential risk factors (age, smoking, duration of type 2 diabetes mellitus, body mass index (BMI), HbA1c, systolic blood pressure (SBP), diastolic blood pressure (DBP), urea, BUN, eGFR, and total cholesterol) of diabetic kidney disease were examined using multivariable regression analysis. The prevalence ratio rather than the odds ratio was estimated for diabetic kidney disease outcome since DKD was prevalent 67% in the study sample. This was done to avoid overestimating the strength of the link, as suggested by the literature [22]. To investigate the relationship between diabetic kidney disease and patient background variables, a Poisson regression utilizing generalized linear model (GLM) with robust variance based on Huber’s sandwich estimator was conducted (i.e., demographic, clinical, and laboratory parameters) [23]. ANGPTL-4 outcomes were converted to natural log scale for better presentation in the Poisson regression. Prevalence ratios (PRs) with 95% confidence intervals (CIs) are used to represent the Poisson regression findings. To ensure validity of the results, we checked co-linearity (where variance inflation factor >10) between covariates in the model, and clinically meaningful interactions by including cross-product terms and were considered significant if *p* < 0.1.

The diagnostic value of ANGPTL-4 and KIM-1 urine levels was evaluated using ROC (receiver operating characteristic) curve analysis. Through the Youden index, the optimal cutoff values were determined and were derived with sensitivity (SN) and specificity (SP). The IBM SPSS 26.0 statistics program for Windows (IBM Corp, 188 Armonk, NY, USA) and Stata software version 17.0 (StataCorp, College Station, TX, USA) were used to conduct the statistical analysis. Two-sided *p*-value < 0.05 was considered statistically significant except for Bonferroni corrected comparisons.

## 3. Results

### 3.1. Study Population Characteristics and Biochemical Parameter Assessment

The study enrolled 135 participants with a mean age of 60.1 ± 10.8 years. They were categorized into three groups, with 45 each in the normoalbuminuria, microalbuminuria, and macroalbuminuria groups. Age, SBP, DBP, total protein, total cholesterol, and electrolytes including Na, K, and Cl did not significantly differ between the three study groups. The patients with macroalbuminuria, as compared to those with normoalbuminuria and microalbuminuria, had a longer duration of T2DM and increased FPG, HbA1c, urea, uric acid, BUN, and serum creatinine. Demographic details and biochemical parameter comparisons are presented in Table 1.

### 3.2. Comparison of Urinary Levels of ANGPTL-4 and KIM-1 among the Different Study Groups

Concentrations of ANGPTL-4, KIM-1, and UACR were conclusively related. Patients in the macroalbuminuria category had a significantly higher value of ANGPTL-4 (3404.7 ± 482 pg/mL) as compared to the patients in the normoalbuminuria and microalbuminuria groups (*p* < 0.001). Further, urinary KIM-1 levels were also positively related to UACR and increased successively. Macroalbuminuria group patients had a significantly elevated value of KIM-1 (3.0 ± 1.3 ng/mL; *p* < 0.001) in comparison to the other two groups (Table 1 and Figure 2A,B). Both the uANGPTL-4 and uKIM-1 levels adjusted by urinary creatinine were elevated in patients in the microalbuminuria group and were further raised in the macroalbuminuria group (Table 1 and Figure 3A,B). Furthermore, the concentrations of ANGPTL-4 and KIM-1 were stratified according to the eGFR levels. Patients in stages IV and V of chronic kidney disease (CKD) had relatively high concentrations of ANGPTL-4 (3487.3 ± 951.1 pg/mL and 3659.9 ± 806. 1 pg/mL, respectively) as compared to the patients in stages I (1775.2 ± 520.7 pg/mL), II (2303.7 ± 717.9 pg/mL), and III (3028.3 ± 720.4 pg/mL) (Figure 4A). Similarly, KIM-1 levels significantly increased from stage I to stage V. Patients falling in stages I (0.0852 ± 0.0624 ng/mL) and II (1.495 ± 0.789 ng/mL) had the lowest concentration, and as the kidney function deteriorated, a steep rise in the KIM-1 levels was observed (stage V: 3.772 ± 1.812 ng/mL) (Figure 4B).

### 3.3. Correlation of KIM-1 and ANGPTL-4 with Clinical Parameters

The urinary concentration of ANGPTL-4 was positively correlated with age, duration of diabetes, FPG, HbA1c, urea, serum creatinine, urine albumin, and UACR in all three groups. Similarly, for urinary KIM-1 levels, a significant positive correlation was found with SBP, HbA1c, urea, BUN, serum creatinine, urine albumin, and UACR. Subsequent Spearman’s correlation revealed that both the urinary biomarkers ANGPTL-4 and KIM-1 were negatively associated with eGFR (ANGPTL-4: r = −0.2043; r = −0.5235; r = −0.8149, *p* < 0.0001; KIM-1: r = −0.3262; r = −0.6820; r = −0.8542, *p* < 0.0001) (Table 2).

### 3.4. Associations between Urinary ANGPTL-4 and KIM-1 and Diabetic Kidney Disease

The urinary levels of ANGPTL-4 and KIM-1 were significantly associated with diabetic kidney disease even after adjustments were made for demographic, clinical, and laboratory parameters (Table 3). Four regression models were fitted with each marker in a hierarchal fashion (model 1: unadjusted; model 2: adjusted for demographic and clinical variables; lab parameters in the model 3; and finally model 4 by adjusting significant factors from model 2 and 3). The PR for both KIM-1 and ANGPTL-4 was statistically significant in each model, indicating its predictive ability. Finally, model 4 showed increasing prevalence of DKD via a per unit increase in KIM-1 (PR: 1.25; 95% CI: 1.14 to 1.38; *p* < 0.001) and ANGPTL-4 (PR: 3.40; 95% CI: 2.32 to 4.98; *p* < 0.001) on the log scale.

### 3.5. Sensitivity and Specificity of ANGPTL-4 and KIM-1 in the Diagnosis of DKD

To assess the diagnostic utility and accuracy of urinary ANGPTL-4 and KIM-1 as individual markers and in the combined form as a panel, ROC analysis was performed (Table 4; Figure 5A,B). In microalbuminuria group, KIM-1 had a relatively large area under curve (AUC) (0.967; 95%CI: 0.928–1.000; *p* < 0.0001) as compared to ANGPTL-4 (0.899; 95%CI: 0.833–0.966; *p* < 0.0001) whereas panel had a similar AUC (0.967; 95%CI: 0.932–1.000; *p* < 0.0001) to that of KIM-1. Furthermore, in the macroalbuminuria group, urinary ANGPTL-4 had a slightly higher AUC as compared to that of urinary KIM-1; the AUC for ANGPTL-4 was 1.000 (95%CI: 1.000–1.000, *p* < 0.0001), and the AUC for KIM-1 was 0.998 (95%CI: 0.993–1.000, *p* < 0.0001), while the panel analysis of the biomarkers yielded a perfect AUC of 1 (95%CI: 1.000–1.000, *p* < 0.0001).

## 4. Discussion

Globally, DKD is the most common reason of end-stage renal disease and is classically presented by glomerular hyperfiltration and albuminuria. However, it is evident that microalbuminuria may not be considered a specific and sensitive predictor of DKD [24]. The prevention of DKD and the retarding of its rate of progression at the earliest time are vital; hence, strengthening the diagnostic and treatment approach necessitates the development of new markers. In the present work, we examined the combined efficacy of podocyte injury marker ANGPTL-4 and tubular injury marker KIM-1 for the initial identification of DKD.

In the current investigation, we demonstrated that the urinary concentrations of both ANGPTL-4 and KIM-1 were elevated in macroalbuminuria patients as compared to the patients in the normoalbuminuria group. Moreover, ANGPTL-4 and KIM-1 levels increased with the deterioration of kidney function, as both the markers were negatively correlated with eGFR. DKD pathogenesis is most likely to occur as a result of the interplay between metabolic factors, such as the accumulation of glycated end products and hemodynamic processes including increased intraglomerular and systemic pressure and various hormone pathways, such as the renin–angiotensin–aldosterone system [25]. These interactions cause various microstructural changes in the kidney that further contribute to renal dysfunction. Podocyte effacement or podocyte-specific mutations are the early pathophysiological developments that lead to albumin precipitation in the urine [26]. Recent studies reported that higher glomerular Angptl4 expression triggered filtration impairment in the glomerular basement membrane, leading to albuminuria; additionally, podocyte-specific transgenic rats at the age of 1 month exhibited proteinuria even when the podocyte foot processes remained normal [14]. The present findings showed that ANGPTL-4 levels in the urine were significantly higher in the macroalbuminuria group than in the microalbuminuria group. A previous study demonstrated that ANGPTL-4 was over-expressed in minimally changed disease and was associated with proteinuria and hyper triglyceridemia [13]. However, another study showed that urinary ANGPTL-4 was elevated in patients with colossal proteinuria despite the underlying renal disease [9]. Further, a clinical study reported that ANGPTL-4 was correlated with chronic kidney disease and may act as a universal marker of declining kidney function [27]. The current study found the urinary levels of ANGPTL-4 to be significantly higher in DKD patients with macroalbuminuria than in the patients with microalbuminuria. Additionally, in line with past studies, it was also discovered that uANGPTL-4 was substantially correlated with FPG, HbA1c, and BUN [9,12]. Furthermore, we identified a negative correlation between ANGPTL-4 and eGFR, indicating the worsening of kidney function on over expression of ANGPTL-4 in the urine. This was further confirmed by the multivariate analysis in which ANGPTL-4 was found to be significantly prevalent in a diabetic kidney disease population even after adjustments were made for potential confounders, including duration of T2DM, HbA1c, BUN, and eGFR. Moreover, the expression of ANGPTL-4 in urine may be an outcome of podocytopathy at an earlier stage of renal dysfunction. Although glomerulus alteration is of paramount consideration in the pathophysiology of diabetic kidney disease, tubulointerstitial injury is equally vital in the disease progression process and serves as a potential predictor for the decline in renal function [28]. The onset of a tubular or glomerular insult is uncertain; thus, it is advisable to consider both potential molecular pathways to halt DKD at an early stage. KIM-1 is a transmembrane protein that is remarkably upregulated in proximal tubule injury, and its ectodomain cleaves and sheds from cells and is easily quantifiable in urine [10]. In our study, KIM-1 levels were significantly correlated with UACR and increased significantly from the normoalbuminuria group to the macroalbuminuria group, similar to what was reported in the study by De Carvalho JA et al. [29], which showed a progressive rise in uKIM-1 levels in T2DM patients with UACR from a normal to severe range. Expression of Kim-1 at the primary stage of DKD may suggest early tubular involvement. Furthermore, Peralta CA et al. [30] reported a higher concentration of urinary KIM-1 independent of albuminuria, recommending the usage of urinary KIM-1 in the detection of DKD and renal dysfunction even at the early stage when microalbuminuria is not evident. KIM-1 was negatively correlated with eGFR, in line with the study conducted by Allgaier R et al. [31] in which there was an inverse relation between eGFR and uKIM-1 values. KIM-1 was also found to be positively associated with HbA1c, urea, BUN, serum creatinine, urine albumin, and UACR. Multivariable analysis showed that urinary KIM-1 was significantly prevalent in patients with diabetic kidney disease even after demographic, lifestyle and clinical risk factors were controlled for. This finding aligns with an 8.7-year follow-up study in which KIM-1 (aHR, 1.17; 95% CI, 1.05 to 1.30) was linked with a higher risk of diabetic kidney disease progression [32].

An ROC curve was plotted to assess the diagnostic performance of urinary ANGPTL-4 and urinary KIM-1, while the Youden index was used for the calculation of optimal cutoff value. Both the parameters were found to perform excellently individually as well as in a combined form. Following the study conducted by Vanarsa K et al. [28] that showed excellent performance of urinary ANGPTL-4 in the diagnosis of lupus nephritis, with an AUC of 0.96 and a specificity of 87.5%, our study also showed a subsequent increase in the AUC from the microalbuminuria to the macroalbuminuria group, and the best performance was found in the macroalbuminuria group with 2794 as the optimal cutoff value and a 97.8% specificity. Similarly, urinary KIM-1 had a significantly large AUC of 0.998 with an optimal cut off value of 1.87 and a specificity of more than 95% in the macroalbuminuria group. Unlike Ding Y et al. [33] who reported an AUC forKIM-1 of less than 0.700 in differentiating tubulointerstitial lesions and hence considered the performance non-ideal, in our study, KIM-1 exhibited a higher AUC value in the microalbuminuria and macroalbuminuria group and hence demonstrated a significantly stronger diagnostic performance. The panel analysis of the two biomarkers in the combined form exhibited a fairly good AUC of 0.967 in the macroalbuminuria group and yielded a perfect AUC of 1 in the macroalbuminuria group, indicating the diagnostic efficacy of the biomarkers when combined.

All the renal compartments are subjected to various hemodynamic and metabolic variables associated with diabetes mellitus, especially hyperglycemia, and glomerular alterations and tubulointerstitial damage may already be initiated even before proteinuria [29]. Identification of the damage at an early stage could potentially de-escalate the prognosis of DKD.

Some limitations to the work presented here should be considered. The study was single center and cross-sectional in design, with a limited patient pool. The comparatively small sample size of the study limited the precision and power to ascertain associations of moderate strength. However, to remove the biases, we analyzed the effect of the biomarkers in two subgroups (albuminuria and eGFR). A longitudinal follow-up investigation of normoalbuminuria T2DM patients with an eGFR in the normal range would be informative. Furthermore, only T2DM control patients were enrolled, and the absence of healthy age-matched controls limited the ability to account for persons without hyperglycemia. Finally, the impact of pharmacological therapy on the candidate biomarkers was also not ruled out.

## 5. Conclusions

This is the first study to examine the combined expression of urinary ANGPTL-4 and urinary Kim-1 together for the detection of DKD in the Asian population. Even after the established risk variables were controlled for, the ANGPTL-4 and KIM-1 concentrations were independently linked with DKD. Both the biomarkers showed good stage-specific sensitivity individually as well as in combination. The study findings suggest that a combination of ANGPTL-4 and KIM-1 may strengthen the diagnostic approach to diabetic kidney disease. However, further longitudinal studies in different large ethnic populations are required to validate the marker’s suitability.

## Figures and Tables

**Figure 1 jpm-13-00577-f001:**
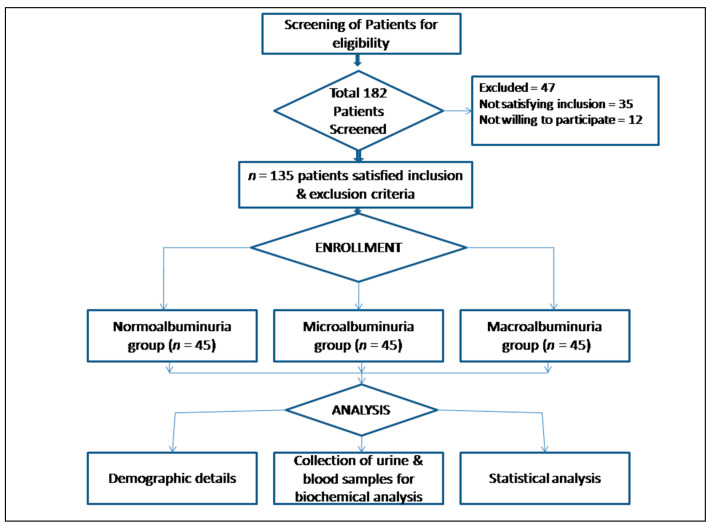
Schematic presentation of flow of participants.

**Figure 2 jpm-13-00577-f002:**
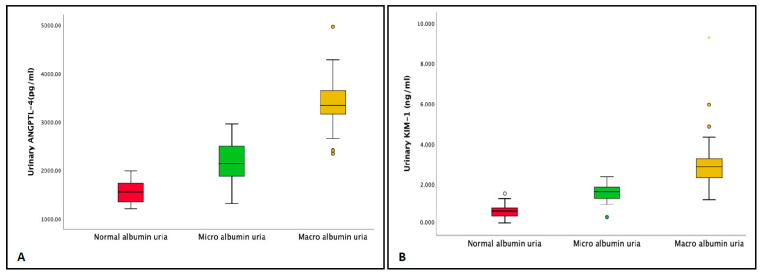
Urinary levels of the biomarkers (**A**) ANGPTL-4 and (**B**) KIM-1 in the patients groups classified according to urinary albumin–creatinine ratio (UACR) levels; normoalbuminuria UACR < 30 mg/g, microalbuminuria UACR 30 to 300 mg/g, and macroalbuminuria UACR > 300 mg/g.

**Figure 3 jpm-13-00577-f003:**
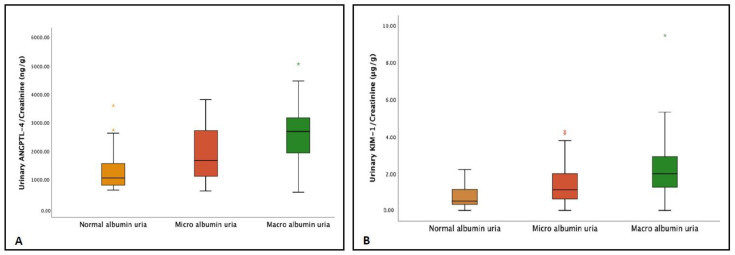
Urinary levels of the biomarkers (**A**) ANGPTL-4 and (**B**) KIM-1 adjusted with urinary creatinine in the patients groups classified according to urinary–creatinine ratio (UACR) levels; normoalbuminuria UACR < 30 mg/g, microalbuminuria UACR 30 to 300 mg/g, and macroalbuminuria UACR > 300 mg/g.

**Figure 4 jpm-13-00577-f004:**
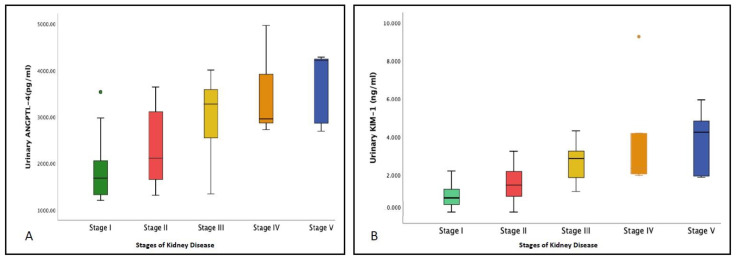
Urinary levels of the biomarkers (**A**) ANGPTL-4 and (**B**) KIM-1 in the patients classified on the basis of estimated glomerular filtration rate (eGFR) values into stage I CKD (eGFR > 90 mL/min/1.73 m^2^), stage II CKD (eGFR: 60–89 mL/min/1.73 m^2^), stage III CKD (eGFR: 30–59 mL/min/1.73 m^2^), stage IV CKD (eGFR: 15–29 mL/min/1.73 m^2^), and stage V CKD (eGFR < 15 mL/min/1.73 m^2^).

**Figure 5 jpm-13-00577-f005:**
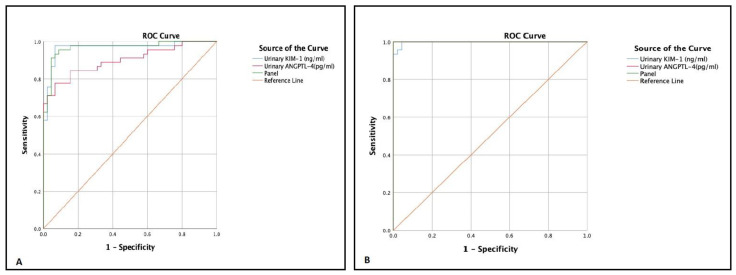
Receiver operating characteristic (ROC) analysis was performed for the prediction of diagnostic performance. (**A**) Type 2 diabetes mellitus patients with microalbuminuria [KIM-1: AUC 0.967; ANGPTL4: AUC 0.899; Panel: AUC 0.967, *p* < 0.0001]. (**B**) Type 2 diabetes mellitus patients with macroalbuminuria [KIM-1: AUC 0.998; ANGPTL4: AUC 1; Panel: AUC 1, *p* < 0.0001].

**Table 1 jpm-13-00577-t001:** Comparison of demographic, clinical, and laboratory parameters between the patient groups.

Background Parameters	Normoalbuminuria (Control Group) (*n* = 45)	Microalbuminuria (*n* = 45)	Macroalbuminuria (*n* = 45)	*p*-Value
Demographic variables				
Age(year)	56.9 ± 8.7	61.9 ± 12.0	61.4 ± 10.9	0.0527
Female gender (%)	23 (51.1)	21 (46.7)	22 (48.9)	0.915
Education (%)				
Illiterate	10 (22.2)	13 (28.9)	11 (24.4)	0.318
Upto Senior Secondary	21 (46.7)	17 (37.8)	24 (53.3)	
Graduation and Above	14 (31.1)	15 (33.3)	7 (15.6)	
Life style and clinical variables				
Smokers (%)	11 (24.4)	13 (28.9)	8 (17.8)	0.459
Family History of T2DM (%)	17 (37.8%)	18 (40%)	17 (37.8%)	
Duration of T2DM (year)	6.3 ± 5.1	10.1 ± 8.6	11.9 ± 6.8	0.0008
BMI (Kg/m^2^)	27.7 ± 3.5	30.1 ± 5.3	30.0 ± 5.5	0.0276
SBP (mmHg)	139.9 ± 13.4	141.6 ± 11.7	137.0 ± 10.8	0.1810
DBP (mmHg)	85.0 ± 6.1	87.1 ± 6.8	84.2 ± 6.0	0.0791
Laboratory variables				
FPG (mg/dL)	152.4 ± 44.6	182.4 ± 53.3	190.1 ± 42.3	0.0005
HbA1c (%)	7.2 ± 0.8	9.4 ± 1.8	9.8 ± 1.1	<0.001
Albumin (g/dL)	4.4 ± 0.3	4.1 ± 0.5	4.0 ± 0.5	0.0001
Globulin (g/dL)	2.9 ± 0.4	2.9 ± 0.3	3.1 ± 0.4	0.0021
Albumin Globulin Ratio	1.6 ± 0.2	1.4 ± 0.2	1.3 ± 0.3	0.0001
Protein Total (g/dL)	7.3 ± 0.5	7.0 ± 0.6	7.1 ± 0.4	0.0382
Urea (mg/dL)	27.1 ± 12.4	34.3 ± 13.4	37.1 ± 10.2	0.0005
BUN (mg/dL)	13.0 ± 6.4	16.1 ± 7.0	17.6 ± 4.7	0.0016
Uric Acid (mg/dL)	4.0 ± 0.9	5.5 ± 0.7	6.4 ± 0.7	<0.001
Serum Creatinine (mg/dL)	0.9 ± 0.1	1.2 ± 0.6	1.3 ± 0.7	0.0002
eGFR (mL/min/1.73 m^2^)	88.7 ± 11.7	68.1 ± 24.8	62.4 ± 22.6	<0.001
Total Cholesterol (mg/dL)	178.5 ± 42.1	198.8 ± 46.4	196.6 ± 46.2	0.0661
Na (mmol/L)	140.3 ± 3.2	140.4 ± 3.1	140.6 ± 3.1	0.8805
K (mmol/L)	4.7 ± 0.6	4.8 ± 0.6	4.9 ± 0.6	0.0428
Cl (mmol/L)	103.3 ± 3.0	103.4 ± 3.3	104.1 ± 2.9	0.3581
Ca (mg/dL)	9.1 ± 0.7	8.6 ± 0.8	8.7 ± 0.8	0.0105
P (mg/dL)	3.5 ± 0.7	3.9 ± 0.7	4.0 ± 0.8	0.0043
Alkaline Phosphatase (U/L)	94.9 ± 13.0	96.3 ± 16.5	104.6 ± 17.8	0.0092
Urine Albumin (mg/L)	10.4 ± 4.9	170.4 ± 81.5	442.9 ± 84.1	<0.001
Urine Creatinine (mg/dL)	167.9 ± 30.5	135.1 ± 29.4	110.5 ± 11.2	<0.001
Urine Albumin/Creatinine (mg/g)	6.7 ± 3.9	131.4 ± 72.0	403.2 ± 81.0	<0.001
Urinary ANGPTL-4 (pg/mL)	1549.6 ± 229.5	2180.5 ± 424.5	3404.7 ± 482.0	<0.001
Urinary KIM-1 (ng/mL)	0.6 ± 0.4	1.6 ± 0.4	3.0 ± 1.3	<0.001
Urinary ANGPTL-4/Creatinine (ng/g)	965.2 ± 282.6	1691.2 ± 511.2	3119.0 ± 600.5	<0.001
Urinary KIM-1/Creatinine (µg/g)	0.4 ± 0.3	1.2 ± 0.4	2.8 ± 1.4	<0.001

Data are presented as mean ± SD unless specified; *p*-value was calculated using either chi-square test or one-way ANOVA or Kruskal–Wallis H test; T2D—type 2 diabetes mellitus; BMI—body mass index; SBP—systolic blood pressure; DBP—diastolic blood pressure; FPG—fasting plasma glucose; HbA1c—glycosylated hemoglobin; BUN—blood urea nitrogen; eGFR—estimated glomerular filtration rate; Na—sodium; K—potassium; Cl—chloride; Ca—calcium; P—phosphorus; ANGPTL-4—angiopoietin-like protein-4; KIM-1—kidney injury molecule-1. Patients groups are classified according to the levels of urinary albumin–creatinine ratio (UACR); normoalbuminuria UACR < 30 mg/g, microalbuminuria UACR 30 to 300 mg/g, macroalbuminuria UACR > 300 mg/g.

**Table 2 jpm-13-00577-t002:** Spearman rank correlation ofANGPTL-4 (pg/mL) and urinary KIM-1 (ng/mL) with different parameters of the study.

Parameters	Urinary ANGPTL-4			Urinary KIM-1		
	Normoalbuminuria	Microalbuminuria	Macroalbuminuria	Normoalbuminuria	Microalbuminuria	Macroalbuminuria
Age (Year)	0.39 *	0.39 *	0.39 *	0.14	0.57 ***	0.45 *
Duration of T2DM (year)	0.36 *	0.48 **	0.36 *	0.02	0.66 ***	0.40 *
BMI (Kg/m^2^)	0.28	0.31 *	0.28	0.19	0.04	0.31 *
SBP (mmHg)	0.21	0.30 *	0.21	0.30 *	0.47 *	0.36 *
DBP (mmHg)	0.26	0.26	0.26	0.36 *	0.32 *	0.27
FPG (mg/dL)	0.42 *	0.33 *	0.42 *	0.18	0.44 *	0.33 *
HbA1c (%)	0.43 *	0.32 *	0.43 *	0.40 *	0.34 *	0.36 *
Albumin (g/dL)	−0.01	0.23	−0.01	0.09	0.07	−0.23
Globulin (g/dL)	0.14	−0.03	0.14	0.64 ***	−0.12	0.36 *
Albumin Globulin Ratio	−0.07	0.14	−0.07	−0.45 *	0.09	−0.30 *
Protein Total (g/dL)	0.16	0.16	0.16	0.55 **	−0.04	0.11
Urea (mg/dL)	0.30 *	0.43 *	0.30 *	0.38 *	0.64 ***	0.39 *
BUN (mg/dL)	0.29	0.31 *	0.29	0.39 *	0.52 **	0.39 *
Uric Acid (mg/dL)	−0.02	0.15	−0.02	−0.18	0.14	−0.02
Serum Creatinine (mg/dL)	0.70 ***	0.53 ***	0.70 ***	−0.03	0.73 ***	0.68 ***
eGFR (mL/min/1.73 m^2^)	−0.20 ***	−0.52 ***	−0.81 ***	−0.33 *	−0.68 ***	−0.85 ***
Total Cholesterol (mg/dL)	0.42 *	0.28	0.42 *	−0.16	0.12	0.60 ***
Urine Albumin (mg/L)	0.80 ***	0.71 ***	0.80 ***	0.32 *	0.82 ***	0.69 ***
Urine Creatinine (mg/dL)	−0.30 *	−0.23	−0.30 *	−0.36 *	−0.24	−0.44 **
Urine Albumin/Creatinine (mg/g)	0.88 ***	0.73 ***	0.97 ***	0.38 *	0.84 ***	0.87 ***
Urinary KIM-1/Creatinine (µg/g)	0.77 ***	0.58 ***	0.77 ***	0.96 ***	0.85 ***	0.97 ***
Urinary ANGPTL-4/Creatinine (ng/g)	0.84 ***	0.76 ***	0.84 ***	0.62 ***	0.60 ***	0.77 ***

T2DM—type 2 diabetes mellitus, BMI—body mass index; SBP—systolic blood pressure; DBP—diastolic blood pressure; FPG—fasting plasma glucose; HbA1c—glycosylated hemoglobin; BUN—blood urea nitrogen; eGFR—estimated glomerular filtration rate; ANGPTL-4—angiopoietin-like protein-4; KIM-1—kidney injury molecule-1.*** Correlation is significant at the 0.0001 level (2-tailed).** Correlation is significant at the 0.001 level (2-tailed). * Correlation is significant at the 0.05 level (2-tailed).

**Table 3 jpm-13-00577-t003:** Association between urinary markers (ANGPTL-4 and KIM-1) and diabetic kidney disease after adjustment for demographic, clinical, and laboratory parameters.

Biomarker	PR	95% CI	*p*-Value
Log [UrinaryANGPTL-4]			
Model 1: unadjusted	4.36	3.04 to 6.24	*p* < 0.001
Model 2: adjusted for demographic and clinical history [age, gender, BMI, family history of T2DM, duration of T2DM]	4.38	2.98 to 6.44	*p* < 0.001
Model 3: adjusted for laboratory parameters [SBP, DBP, HbA1C, urea, BUN, serum creatinine, eGFR, total cholesterol]	3.08	2.00 to 4.75	*p* < 0.001
Model 4: adjusted for significant variables in model 2 and model 3 [HbA1C, serum creatinine]	3.40	2.32 to 4.98	*p* < 0.001
Urinary KIM-1			
Model 1: unadjusted	1.26	1.12 to 1.42	*p* < 0.001
Model 2: adjusted for demographic and clinical history [age, gender, BMI, family history of T2DM, duration of T2DM]	1.30	1.18 to 1.43	*p* < 0.001
Model 3: adjusted for laboratory parameters [SBP, DBP, HbA1C, urea, BUN, serum creatinine, eGFR, total cholesterol]	1.17	1.04 to 1.31	0.008
Model 4: adjusted for significant variables in model 2 and model 3 [age, BMI, duration of T2DM, HbA1C, serum creatinine]	1.25	1.14 to 1.38	*p* < 0.001

PR—prevalence ratio; CI—confidence interval; BMI—body mass index; T2DM—type 2 diabetes mellitus; HbA1c—glycosylated hemoglobin; SBP—systolic blood pressure; DBP—diastolic blood pressure; BUN—blood urea nitrogen; eGFR—estimated glomerular filtration rate; ANGPTL-4—angiopoietin-like protein-4; KIM-1—kidney injury molecule-1. *p*-value < 0.001 statistically significant.

**Table 4 jpm-13-00577-t004:** ROC analysis for urinary ANGPTL-4 and urinary KIM-1.

Test Result Variable(s)	Optimal Cutoff	SN	SP	Positive Likelihood Ratio	AUC	Asymptotic 95% Confidence Interval		*p*-Value
						Lower	Upper	
Urinary ANGPTL-4 (pg/mL)								
Microalbuminuria	1868	77.8%	93.3%	11.61	0.90	0.83	0.97	<0.0001
Macroalbuminuria	2794	100%	97.8%	45.45	1.00	1.00	1.00	<0.0001
Urinary KIM-1 (ng/mL)								
Microalbuminuria	0.96	97.8%	93.3%	14.59	0.97	0.93	1.00	<0.0001
Macroalbuminuria	1.87	100%	95.6%	22.72	0.98	0.95	1.00	<0.0001

SN—sensitivity; SP—specificity; AUC—area under curve; ANGPTL-4—angiopoietin-like protein-4; KIM-1—kidney injury molecule-1. *p*-value < 0.0001 statistically significant.

## Data Availability

The data presented in this study are available on request from the corresponding author.

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
