# Peer review of "Expression of Angiopoetin-Like Protein-4 and Kidney Injury Molecule-1 as Preliminary Diagnostic Markers for Diabetes-Related Kidney Disease: A Single Center-Based Cross-Sectional Study"

_jpm, 2023, doi:10.3390/jpm13040577_

Round 1
Reviewer 1 Report
General comments
This manuscript by Bano et al. focuses on ANGPTL-4 and KIM-1 as biomarkers of Diabetes-related kidney damage. The topic addressed is interesting. I have annotated the manuscript with several minor corrections, which I believe will improve the readability of the paper.
Specific comments
Line 114 and 115 - Authors write -800 C. Should this be -80.0 °C?
Line 131 and Table 1 - The symbol for phosphorus is P, not Ph.
In Tables 1 and 2 it would be appropriate to add a unit of measurement to "Duration of T2DM".
Grammatical errors are there, Eg: line 194. Commas and periods are missing.
Author Response
Response to Reviewer 1 Comments
We thank the reviewer for his very positive evaluation and his constructive comments. We have tried our best to conduct additional analyses and to improve the presentation of this manuscript. We have included the response to the comments for your reference.
Point 1: Line 114 and 115 - Authors write -800 C. Should this be -80.0 °C?
Response 1: Thank you for your observation. This has been rectified and the manuscript is now updated accordingly.
Point 2: Line 131 and Table 1 - The symbol for phosphorus is P, not Ph.
Response 2: We appreciate your observation and valuable comment. We have now corrected the symbol.
Point 3: In Tables 1 and 2 it would be appropriate to add a unit of measurement to "Duration of T2DM".
Response 3: We have now added the unit to "Duration of T2DM" measurement as per your suggestion.
Point 4: Grammatical errors are there, Eg: line 194. Commas and periods are missing.
Response 4: We thank the reviewer for pointing all these grammatical issues out. We have now rectified the errors and updated the manuscript.
We hope you will find the revised version of the manuscript appropriate for publication.
Please see the attachment for the updated manuscript with suggested corrections.

Reviewer 2 Report
Bano et al. estimated the levels of ANGPTL-4 and KIM-1 in 135 patient samples, and suggsted that these two genes can be used as the preliminary diagnostic markers for DKD. This is a well-done study. But I have some doubts about some results.
1. All clinical data are too good to be true, especially the ROC data. I apologize for my doubts about this part of the data. I suggested the authros could provide the raw data for review.
2. These two genes have been reported as biomarkers for kidney disease. Of course, it has not been reported to use them together, however, I doubt whether the expression of these two genes can distinguish DKD from other kidney diseases, I hope the author can provide data.
3. I suggested the the author can elaborate more on the mechanism of these two genes on DKD in the discussion section.
To sum up, if true they would be nice biomarkers for DKD. I suggested the authors provide the raw data of all their clincal table and figures for the review.
Author Response
Response to Reviewer 2 Comments
We thank the reviewer for his very positive evaluation and his constructive comments. We have tried our best to conduct additional analyses and to improve the presentation of this manuscript. We have included the response to the comments for your reference.
Point 1: All clinical data are too good to be true, especially the ROC data. I apologize for my doubts about this part of the data. I suggested the authros could provide the raw data for review.
Response 1: We sincerely appreciate the recognition and constructive feedback. We have provided the raw data for your reference in order to dispel any question.
Point 2: These two genes have been reported as biomarkers for kidney disease. Of course, it has not been reported to use them together, however, I doubt whether the expression of these two genes can distinguish DKD from other kidney diseases, I hope the author can provide data.
Response 2: We are thankful for this comment, as it points to an important aspect of this study. The presented study aimed to investigate the potential of ANGPTL4 and KIM-1 for the early detection of diabetic kidney disease, as the available markers like microalbuminuria and serum creatinine possess potential limitations in terms of sensitivity and predictive value.
In the presented study, type 2 diabetes mellitus (T2DM) and established diabetic kidney disease (DKD) patients were enrolled. Patients with other renal or urinary tract illnesses confirmed by clinical or laboratory evidences, cerebrovascular disease, inflammatory diseases, infectious disease, cancer, tumors, or recent surgery, were excluded from the study. The results demonstrated that urinary levels of ANGPTL-4 and KIM-1 were increased progressively in T2DM patients from the normoalbuminuria group to the DKD patients with macroalbuminuria. Expression of both markers in T2DM patients with UACR in the normal or slightly elevated range shows a promising role in the early detection of diabetic kidney disease. To further verify the potential roles of ANGPTL-4 and KIM-1 in DKD, correlation analyses and multivariable regression analyses were performed. We found that the urinary ANGPTL-4 and KIM-1 levels were strongly correlated with UACR and eGFR and had fair predictive ability. Also, the prevalence of DKD increased with the per-unit increase in KIM-1 and ANGPTL-4, implying that these two may be a potential factor for evaluating DKD in the future.
Biomarker suitable for monitoring of CKD ought to have narrow biological variability in order to improve the assessment of longitudinal changes. Moreover, it should not be influenced by age, nutrition status or concurrent health concerns. In present study, both the selected biomarkers have proven to be highly sensitive, specific for renal diseases.
According to Chugh SS, et al.1 the discovery of Angptl4 was a major player in human nephrotic syndrome was based on a strategy to identify and selectively investigate genes/ proteins that could potentially link at least two of the three major components (proteinuria, hyperlipidemia, and edema) of nephrotic syndrome.
Li JS, et al.2 reported that podocyte-secreted angiopoietin-like-4 (Angptl4) mediates proteinuria in different types of podocytopathy. In an experimental minimal change disease (MCD) rat model and confirmed MCD patients, they demonstrate that Angptl4 can predict podocyte injury at earlier stages in MCD and the identification of earlier podocyte injury biomarkers could facilitate the prompt diagnosis and treatment of patients with podocytopathy.
Jia S, et al.3 studied biopsy-proven idiopathic immunoglobulin A nephropathy (IgAN) patients and findings showed that Angptl4 levels in plasma and urine were related to podocyte damage and, therefore, might be a promising tool for assessing the severity of IgAN patients to identify and reverse the progression to ESRD.
KIM-1 is not detectable if the kidneys are normal and hence it is a specific and sensitive biomarker for proximal tubule damage.4 Karmakova ТА, et al.5 in their review studied the predicting properties of KIM-1 as an urinary and serologic marker in some kinds of acute and chronic kidney injury, renal cell carcinoma, cardiovascular diseases. Increase of KIM-1 expression and an elevated uKIM-1 level are described in focal glomerulosclerosis, proliferative and membrane glomerulonephritis, IgA nephropathy, diabetic and hypertensive nephropathy, chronic allograft nephropathy, lupus nephritis, etc.
- Chugh SS, Mace C, Clement LC, Del Nogal Avila M, Marshall CB. Angiopoietin-like 4 based therapeutics for proteinuria and kidney disease. Front Pharmacol. 2014;5:23.
- Li JS, Chen X, Peng L, et al. Angiopoietin-Like-4, a Potential Target of Tacrolimus, Predicts Earlier Podocyte Injury in Minimal Change Disease. PLoS One. 2015;10(9):e0137049.
- Jia S, Peng X, Liang L, et al. The Study of Angptl4-Modulated Podocyte Injury in IgA Nephropathy. Front Physiol. 2021;11:575722.
- van Timmeren MM, van den Heuvel MC, Bailly V, Bakker SJ, van Goor H, Stegeman CA. Tubular kidney injury molecule-1 (KIM-1) in human renal disease. J Pathol. 2007;212(2):209-217.
- Karmakova ТА, Sergeeva NS, Kanukoev КY, Alekseev BY, Kaprin АD. Kidney Injury Molecule 1 (KIM-1): a Multifunctional Glycoprotein and Biological Marker (Review). Sovrem Tekhnologii Med. 2021;13(3):64-78. doi:10.17691/stm2021.13.3.08
Point 3: I suggested the the author can elaborate more on the mechanism of these two genes on DKD in the discussion section.
Response 3: Thank you for your valuable input. We have updated the discussion section on the mechanism of the two genes.
We hope you will find the revised version of the manuscript appropriate for publication.
Please see the attachment for the revised manuscript as per the suggestion. The raw data is forwarded to the editor as there is no scope to upload the excel file here.

Round 2
Reviewer 2 Report
The author provided detailed raw data to solve all my doubts.